# One dimensional approximations of neuronal dynamics reveal computational strategy

**Connor Brennan** [1], **Adeeti Aggarwal** [1], **Rui Pei** [2], **David Sussillo** [3,4], **Alex Proekt** [5] *

**1** Department of Neuroscience, University of Pennsylvania, Philadelphia, Pennsylvania, United States of America, **2** Department of Psychology, Stanford University, Palo Alto, California, United States of America, **3** Stanford Neurosciences Institute, Stanford University, Palo Alto, California, United States of America, **4** Department of Electrical Engineering, Stanford University, Palo Alto, California, United States of America, **5** Department of Anesthesiology and Critical Care, University of Pennsylvania, Philadelphia, Pennsylvania, United States of America

☉ These authors contributed equally to this work.
* proekt@gmail.com

**Data Availability Statement:** Data and code used for this work are available at: https://osf.io/bge2n/ The most up to date version of LOOPER is available at: https://github.com/proektlab/LOOPER.

## Abstract

The relationship between neuronal activity and computations embodied by it remains an open question. We develop a novel methodology that condenses observed neuronal activity into a quantitatively accurate, simple, and interpretable model and validate it on diverse systems and scales from single neurons in *C. elegans* to fMRI in humans. The model treats neuronal activity as collections of interlocking 1-dimensional trajectories. Despite their simplicity, these models accurately predict future neuronal activity and future decisions made by human participants. Moreover, the structure formed by interconnected trajectories—a scaffold—is closely related to the computational strategy of the system. We use these scaffolds to compare the computational strategy of primates and artificial systems trained on the same task to identify specific conditions under which the artificial agent learns the same strategy as the primate. The computational strategy extracted using our methodology predicts specific errors on novel stimuli. These results show that our methodology is a powerful tool for studying the relationship between computation and neuronal activity across diverse systems.

## Author summary

Advances in neuronal imaging techniques now allow for the recording of an appreciable fraction of neurons in biological systems. However, it is not clear how to extract scientific insight from such complex nonlinear time series data. Here we develop a methodology for the direct interpretation of the computational algorithms used by several distinct biological systems during a variety of laboratory tasks. The primary intuition of this methodology is that in order to store information in noisy dynamical systems, the information must be protected against noise. This criterion results in systems that can be well approximated by one-dimensional trajectories. These trajectories map directly to the computation performed by the system, and don't dramatically reduce the quantitative accuracy of the models. We apply this method to several different biological systems and artificial neural

**Funding:** This work receive funding from the National Institute of Neurological Disorders and Stroke (R01NS113366 to AP) and through Google (PhD Fellowship to CB). The funders had no role in study design, data collection and analysis, decision to publish, or preparation of the manuscript.

**Competing interests:** The authors have declared that no competing interests exist.

networks. We find that biological systems tend to find generalistic solutions to problems, while artificial neural networks rely on specifics of the dataset.

## Introduction

One of the most ubiquitous approaches to understanding how the brain represents, stores, and processes information is to identify neuronal correlates of environmental conditions and behavioral actions. It is not a priori obvious, however, at which level these correlations should be established. Classical approaches focused on the level of individual neurons. Indeed, firing of individual neurons can be correlated with features of sensory stimuli [1], location of the animal in space [2, 3], and many others. However, in many systems, the activity of individual neurons does not have an obvious relationship with the environment, nor task-relevant variables [4, 5]. Activity of each neuron may be correlated with a complex mixture of salient environmental conditions, vary from individual to individual [6], and across time within the same individual [7]. Interestingly, in systems such as those that regulate motor behaviors [8, 9], cognitive tasks [5, 10] and encoding of sensory stimuli [11], it has been shown that the complex and variable behavior of individual neurons can co-exist with simple collective dynamics of neuronal populations that have a clear relationship to specific task features. These observations spurred a new theory called computation through dynamics (CTD) that attempts to relate neuronal population dynamics to the computations carried out by the brain in service of a task [12].

Rather than examining activity of individual neurons, CTD casts the brain as a dynamical system of the form:

$$\dot{\mathbf{x}}(t) = f(\mathbf{x}(t), \mathbf{u}(t)) \tag{1}$$

The flow of the system, $\dot{\mathbf{x}}$, is uniquely determined by its current state $\mathbf{x}$, and the inputs to the system, $\mathbf{u}$. Thus, the explanatory focus of CTD is not on individual neurons, but on the structure of the trajectories traced by neuronal population activity during a behavioral task. However, in contrast to the activity of individual neurons, which can be directly observed experimentally, extracting neuronal population trajectories from experimental recordings faces several fundamental challenges.

The traditional approach to extracting neuronal population trajectories is to first reduce the dimensionality of the recordings [13, 14] and then identify trajectories in this low dimensional projection [4, 5, 14, 15]. However, it is not clear which dimensions of $\mathbf{x}$ are required to capture the salient features of the trajectories. For instance, to extract rotational trajectories of neuronal population activity in the motor cortex, a projection technique that specifically identifies rotatory dynamics [8] was developed. This required correct *a priori* intuition about the structure of the trajectories that one expects to find. However, such *a priori* intuition may not always be readily gleamed from experimental observations. Furthermore, in some settings neuronal population trajectories are not confined to a single low dimensional subspace but evolve along different dimensions at different locations in state space [16]. In this case, any single linear coordinate system may emphasize some features of the trajectories but obscure others. Thus, in order to make progress towards understanding the computational role of neuronal dynamics, a methodology capable of extracting neuronal population trajectories of arbitrary shape needs to be developed.

Here we present an approach that allows for identification of neuronal population trajectories without making strong assumptions about their shape. Rather than first projecting

neuronal activity onto a single linear subspace, our methodology uses local properties of the dynamics to reconstruct the nonlinear trajectories. The key observation that motivates our approach is that biological systems are noisy [17–20]. Dynamics of noisy dynamical systems can be expressed formally by adding noise to Eq 1.

$$\dot{\mathbf{x}}(t) = f(\mathbf{x}(t), \mathbf{u}(t)) + \epsilon \tag{2}$$

In noisy systems, information stored in the initial position, $\mathbf{x}(0)$, degrades over time due to diffusion [17, 21, 22]. In order to counteract this diffusion and allow for stable information storage two properties—"divergence" and "tangling"—must be minimized [9, 23]. Divergence describes the rate at which points starting from the same initial condition diffuse away from each other. In a system with low divergence, points starting from the same initial condition will remain close together in time—forming a stable trajectory [24, 25]. Stable trajectories occur when all directions in the local neighborhood orthogonal to the flow of the trajectory push the state of the system back to the trajectory, making it robust against local perturbations due to noise. Note that such trajectories will inevitably be one-dimensional, and cannot be combined to form higher dimensional structures (such as sheets), because these structures contain marginally stable directions orthogonal to the flow. When nearby initial conditions have different directions of flow, the trajectories starting from them will tangle. Minimal tangling occurs when stable trajectories are separated in state space and are thus unlikely to mix under the influence of noise.

These theoretical considerations have been supported by empirical investigations showing that neuronal dynamics in cortical areas involved in movement planning and execution specifically avoid divergence and tangling [9, 23]. The natural consequence of minimizing divergence and tangling is that the neuronal dynamics can be well approximated as a collection of stable trajectories that do not overlap. This means that each trajectory uniquely identifies an initial condition and allows for stable information storage. The key insight is that both divergence and tangling are properties defined for a local neighborhood rather than for the entire state space. Our approach, called LOOPER, takes advantage of this and identifies a local low dimensional projection of the system for each neighborhood such that divergence and tangling are minimized. The ultimate result of this modeling approach is that neuronal dynamics are succinctly approximated as a collection of interlocking one-dimensional trajectories.

This approach has several important properties that lend itself to CTD. Because the approach is designed to identify local low dimensional projections that minimize divergence and tangling, the trajectories identified by the method are readily distinguished by the system itself rather than those that can, in principle, be distinguished by a specific choice of the coordinate system imposed by the experimenter. By approximating the dynamics as distinct trajectories, we can directly interrogate the information stored in the system and how that information is processed. Task-salient information will cause the system to have a specific set of initial conditions, which will then evolve in time along a single trajectory. Thus, trajectories are able to store task-salient information. As the system makes decisions with this information, or receives new inputs from the environment, the trajectories can branch or merge. The pattern of connections between the trajectories directly reflects the inferences the system makes on that information—the computational scaffold. Because this scaffold abstracts away the details of the neuronal activity, similar computational scaffolds are extracted from architecturally distinct systems involved in similar computations. Surprisingly, the assumption that neuronal dynamics are composed entirely of stable trajectories does not negatively impact LOOPER's ability to quantitatively model neuronal dynamics. We show that LOOPER faithfully reconstructs experimental observations across scales from single neurons to whole brain

BOLD signals, and across systems of varying complexities from *C. elegans* to the primate prefrontal cortex.

## Results

### 1 Overview of LOOPER

Here we give an intuitive overview of LOOPER. A more detailed description is given in the Methods section. The basic aim of LOOPER is to construct a simple model of complex experimental observations. In order to do this, two fundamental problems need to be solved. First, the experimental observations need to be appropriately simplified. Second, a model of the dynamics of simplified experimental observations needs to be constructed. Remarkably, both problems can be simultaneously addressed for a stochastic dynamical system (Eq 2) by a single mathematical technique called diffusion mapping [26, 27]. The major conceptual innovation made by LOOPER is that, under the assumption that a stochastic dynamical system has been evolved or optimized to store information, models built using diffusion maps can be simplified to a system of interlocking one dimensional stable trajectories.

LOOPER extends a similar methodology used in *C. elegans* [6]. The *C. elegans* methodology used eigendecomposition of diffusion maps to find the primary cyclic mode in the data. However, we find that this technique fails in more complex systems. Eigendecomposition of the diffusion map only finds cyclic trajectories that are associated with a single phase velocity. While this strategy succeeded in *C. elegans*, in more complex systems this may not necessarily be the case. LOOPER addresses this concern by directly clustering the diffusion map to arrive at a model of one-dimensional trajectories. While these clustering and minimization steps inevitably add a level of complexity to the method and keep the method from guaranteeing any sort of optimality, we find that the method is effective at dealing with noisy experimental observations across a wide variety of model organisms and imaging types. Another important departure from the *C. elegans* data is that LOOPER is equipped to deal with trial data. Both LOOPER and the *C. elegans* method are naturally designed to work on continuous recordings, but with a simple modification, (S1 Text section **Extending LOOPER to work with trial data**) the method can also be applied to trial data where continuous recordings are not available.

Due to the stochastic nature of the dynamics it is not possible to precisely model the time-evolution of any one trajectory starting from a particular initial condition. The standard approach to such problems is to replace the stochastic differential equation governing the state of the system (Eq 2) by a deterministic differential equation that governs the time evolution of the distribution of states $P(\mathbf{x}, t)$, known as the Fokker-Planck equation [28]. After appropriate normalization, diffusion maps [26, 27] are discrete approximations of the Fokker-Planck operator. Diffusion maps are also a powerful nonlinear dimensionality reduction technique that can be used for manifold discovery. Thus, diffusion maps are a natural choice for studying dynamics in stochastic systems.

We will illustrate how LOOPER makes use of diffusion mapping to simultaneously reduce the dimensionality of neuronal activity and to construct a model of the dynamics. LOOPER starts from observed neuronal activity time series (Fig 1A). Fig 1B illustrates that the helical structure containing neuronal trajectories (Fig 1A) can be unwrapped to reveal a simpler (lower dimensional) manifold defined by two latent variables using a diffusion map.

In contrast to the traditional approaches [3] that seek to uncover the manifold itself, we are interested in identifying stable neuronal trajectories that lie on the manifold. Note that diffusion maps can be normalized such that each entry is the probability that a data point observed

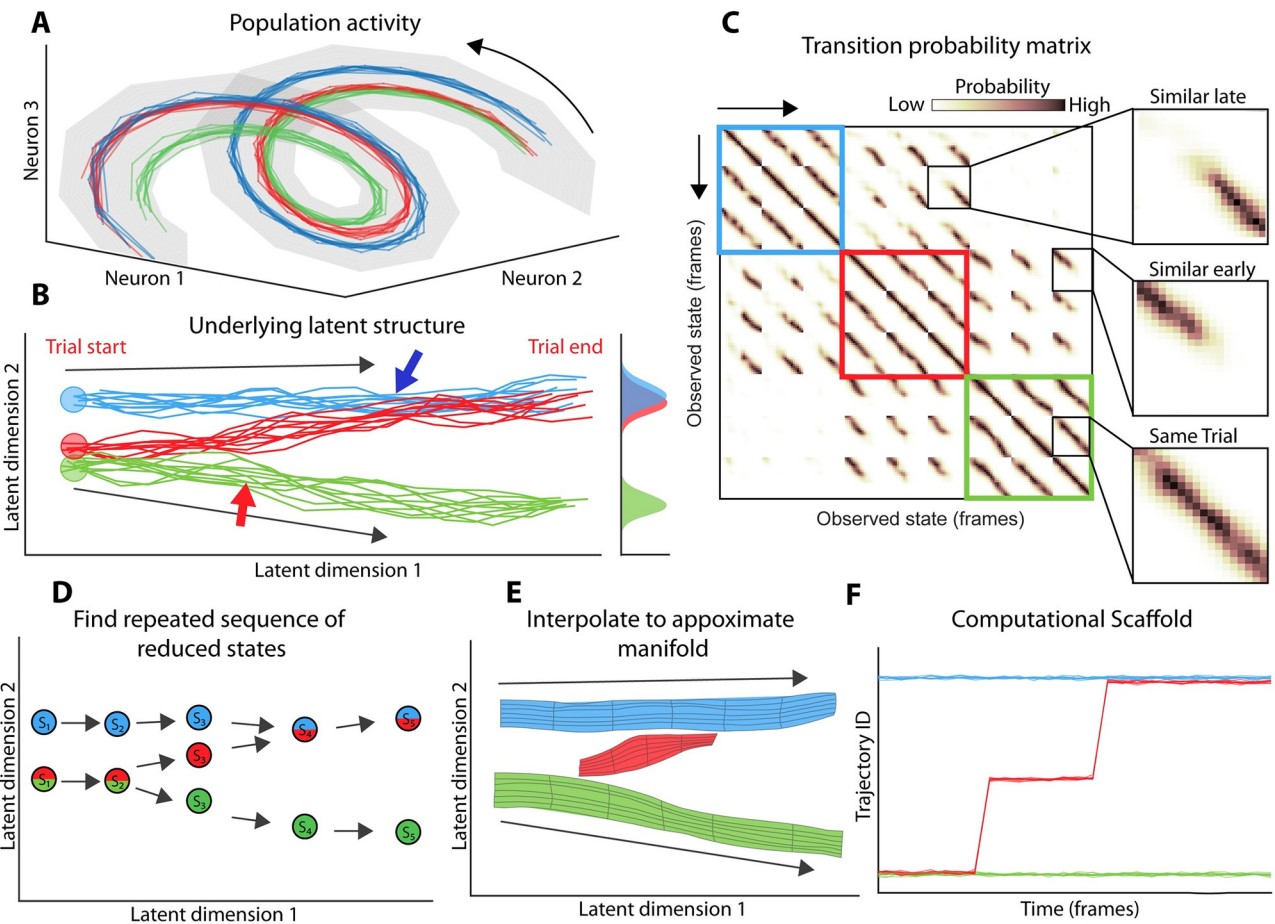

**Fig 1. Schematic of LOOPER—A method for constructing simple models of nonlinear dynamical systems based on experimental observations.**
**Overview**: **A)** Activity of a hypothetical system given by three neurons. The state of this system is characterized by the neuronal population activity vector, which records the instantaneous firing of all neurons (Neurons 1, 2 and 3). Note that the activity is confined to a helical manifold. Time is implicit and flows in the direction indicated by arrows in **A** through **F**. In this example, neuronal activity is recorded under three different task conditions shown by color in **A-F**. **B)** Diffusion mapping is used to unwrap the helical structure in **A** to reveal a simpler (2D) projection that contains all of the neuronal activity. While the unwrapped space is 2-dimensional, the actual dynamics of the system are largely confined to a set of three one dimensional trajectories each starting from a unique set of initial conditions (red, green and blue circles). So long as trajectories stay separate, the information stored in the state of the system is preserved. At the moment when red and blue trajectories merge, the system is no longer able to distinguish them (blue arrow). The merging of the trajectories can be quantified by the overlap in the distributions of neuronal activity (marginal histograms). On the contrary, the green trajectory is separated from the rest at the end of the trial. Thus, the distribution of the green points is distinct from the other two. **C)** Shows a diffusion map, normalized as a transition probability matrix, constructed from the observations in **A**. The three task conditions (three trials of each) are shown by three colored squares. The most likely transitions in each case are along the one off diagonal of the matrix. This corresponds to the actual observations. High transition probabilities away from this main diagonal correspond to similar dynamics observed on different trials. Note that the off diagonal bands appear late in the trials for blue and red conditions. In contrast, the green and red conditions are initially similar but diverge towards the end of the trial. **D)** Observations in **C** naturally inform a way to coarse grain the diffusion map. States characterized by similar transition probabilities (rows of the matrix in **C**) can be readily aggregated. This results in a course grained transition probability matrix where each entry is a cluster of experimental observations aggregated on the basis of similarity of rows in the diffusion map in **C**. This coarse grained transition probability matrix is shown as a graph. The position of each state (S1, S2, S2. . .) in the latent space is given by the mean of all experimental observations that comprise the state. Note that in this example transitions from each state (arrows) are predominantly to a single other state. This expresses the assumption that the most salient information can be represented by one dimensional trajectories. This assumption allows further simplification of the system by clustering states (S1, S2, . . .) into trajectories (ordered sequences of states). **E)** Finally, states within each trajectory are interpolated to reveal a model that consists entirely of one dimensional trajectoriess (each colored surface is a distinct trajectory). The width of each trajectory shows the variance of the experimental observations projected onto the latent space **F)** A useful way to describe the data is the "computational scaffold" which is the trajectory ID assigned to the observed data at each time point. Note that the data tends to separate out the initial conditions (red, green and blue) during the middle of the trial based on trajectory ID. Further, the way that the trajectories merge and split can give valuable insight into the types of information used by the system.

at time $t$ will transition to any other observed data point in the next time step. Thus, the map can be thought of as a transition probability matrix (Fig 1C). This transition probability matrix is a Markov model of the observed dynamics. The major contribution of LOOPER is a methodology to condense diffusion maps to a set of stable one dimensional trajectories in order to arrive at an interpretable and generalizable model of the dynamics.

Note that stable trajectories are readily identifiable in the transition probability matrix as diagonal bands of high transition probability. The most prominent band is on the one-off diagonal, which corresponds to the observed trajectory of the system. Repeated instances of similar trajectories appear as bands in the off diagonals of the matrix (Fig 1C, single trial). This observation can be used to coarse grain the transition probability matrix. Specifically, two states that have similar transition probability distributions (rows of the matrix in Fig 1C) cannot be readily distinguished by the system after a single time step (Methods). Equipped with this natural definition of similarity, LOOPER agglomerates the states of the original diffusion map (Fig 1C) into clusters. The number of clusters is optimized using minimal description length (MDL) algorithm. The result of this clustering procedure is a smaller transition probability matrix (Fig 1D). Note that at the point where the blue and red trajectories cross and tangle (Fig 1C) the clustering procedure leads to convergence of transition probabilities (Fig 1D). Conversely, when the red and the green trajectories separate (Fig 1C) the clustering procedure leads to a divergence of transition probabilities (Fig 1D).

In a system that has stable trajectories, transitions observed in Fig 1D are sparse. Most commonly, a given state can transition to just one other state. Thus, if a system starts out in a particular state at the beginning of a trial, it will most likely emit a stereotyped sequence of states. LOOPER takes advantage of this to further simplify the model. Specifically, we use an additional clustering step to identify repeated state sequences using a modified edit distance. To approximate a continuous trajectory we interpolate between states that comprise it (Fig 1E). Much like in the original coarse graining, the number of trajectories is optimized to preserve the quantitative accuracy of the model.

The net result of this procedure is a simple model of the observed neuronal dynamics defined by just two parameters: trajectory ID and phase along the trajectory. Note that each bin in the model space is associated with a list of experimental observations assigned to it by the clustering algorithm. To validate the model, we project from the highly condensed model space back to neuronal activity space. This is accomplished by emitting the mean of the observed neuronal activity in each bin of the model.

Fig 1E shows how the stable trajectories (colors) of the LOOPER model map onto the latent space. As we will show below for several experimental and model systems a more abstract description of the data is particularly useful (Fig 1F). Rather than visualizing the location of each trajectory in the latent space, we map every experimental observation (marked by a specific time during the task) onto the trajectory ID assigned by LOOPER. We refer to this description as the computational scaffold. This plot contains several fundamental pieces of information. First, it shows how many distinct pieces of information (trial conditions) are stored by the system. In the case of the example in Fig 1, the system distinguishes three trial conditions. It also shows when, during the task, these distinctions are made by the system (point where green and red trajectories diverge). Finally, it shows when the information about different initial conditions is lost (point where blue and red trajectories converge). In this way, the computational scaffold description captures the key information that is stored in the system dynamics. In the rest of the paper, we will show how the computational scaffold can be used to infer both the similarities and the differences in computations embodied in the dynamics of architecturally distinct neuronal networks.

## 2 The computational scaffold extracted by LOOPER predicts task performance on novel stimuli

There is considerable interest in using recurrent neural networks (RNNs) to model dynamics observed in the brain [5, 14, 29, 30]. Regardless of the potential implications for neuroscience, RNNs can approximate any dynamical system [31] with arbitrary accuracy. Therefore, we first illustrate the capabilities of LOOPER on an RNN trained to solve a working memory task. Because we are ultimately interested in modeling noisy brain dynamics, we injected noise into hidden state of the RNN.

In Fig 1, we illustrated the capabilities of LOOPER on a toy system that was explicitly constructed to have stable trajectories. The LOOPER model (Fig 2) successfully reconstructs the dynamics in a noisy RNN ($R^2$ = 0.99). In contrast to the trained RNN, where LOOPER finds discrete stable trajectories, no trajectories are found in an untrained RNN (Fig A in S1 Text). Traditional approaches to studying RNN dynamics involve linear stability analysis [32] to reveal stable fixed points. While this approach is extremely useful for RNNs, it cannot be directly applied to experimental observations. However, in Fig B in S1 Text we show that stable trajectories identified by LOOPER closely correspond to stable fixed points found using conventional stability analysis in the RNN. Thus, LOOPER allows for the discovery of dynamical features similar to those found by linear stability analysis even in biological systems in which the equations of motion are unknown.

The primary advantage of LOOPER over other techniques used to model complex dynamical systems [14, 33, 34] is that the computational scaffold (Fig 2B) can be used to infer the computations implemented in the dynamics. To illustrate how this can be accomplished, we trained the RNN on a working memory task. A schematic of the task is shown in Fig 2A. We picked values of F1 and F2 that mimic the statistics of the values from the originally published data of the same working memory task in rhesus macaques [13, 35].

The goal of the task design is to remember the value of F1 and compare it to the value of F2 after a delay. One strategy that a network can use to solve this task is to remember the values of all three F1s and compare those stored values to F2. However, with the specific values of F1 and F2 used to train the primates, an alternative solution is also possible. Under some task conditions, it is possible to correctly solve the task without any information concerning F1. If F2 happens to be 25, for instance, the correct answer is always "greater than". Similarly, when F2 is 30, the correct response is always "less than". Accordingly, the network may choose to disregard the information concerning F1 and produce the correct response solely on the basis of F2. Note that it is not possible to directly infer which strategy is being used based on responses to the training stimuli.

When we trained an RNN on this stimulus set, LOOPER extracted a computational scaffold which suggests that the two trajectories corresponding to F1 = 20 and F1 = 40 fused before the onset of F2. This implies that while the network originally uniquely encodes each F1, the distinction between F1 = 20 and F1 = 40 is forgotten during the interstimulus period. The other trajectory unequivocally identifies F1 = 10 over the entire interstimulus period. This scaffold suggests the network employs two distinct strategies depending on the specific value of F1. If F1 = 10, then the computational strategy is to compare F1 to F2. However, if F1 is 20 or 40, the network produces a response that depends solely on F2. In some training runs, another computational scaffold which merges F1 = 10 and F1 = 40 was also observed. Both computational scaffolds lead to identical near perfect performance on the training set.

If the computational scaffold uncovered by LOOPER does indeed reflect the information stored and used by the RNN, then the RNN should make specific errors when presented with specific novel combinations of stimuli. Because, according to the LOOPER computational

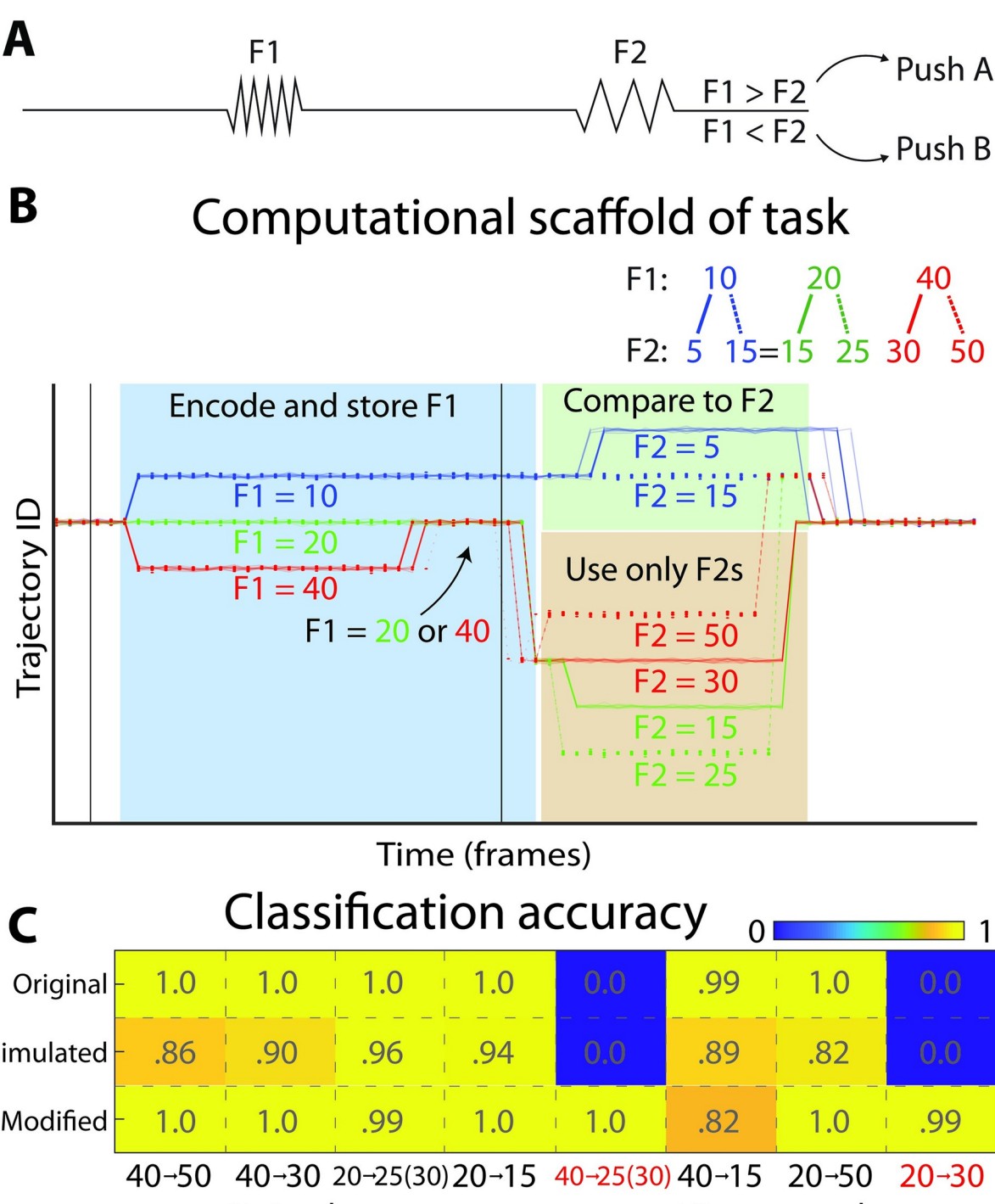

**Fig 2. LOOPER model makes specific predictions about behaviour on novel stimuli combinations. A)** Schematic of the working memory task. We model our task on the Romo task [35]. In their task the monkey receives a sequence of two vibrational stimuli applied to the fingertip (F1 and F2), with an interstimulus delay of 3s. The monkey must push one of the two buttons depending on whether the frequency of F1 is greater than F2, or not. We trained an RNN consisting of 100 LSTM units to solve this task using the same statistics of stimuli as found in the original paper. For the RNN, stimuli are presented for 10 frames. Gaussian noise was added to both the input values, and the hidden states of the RNN. In order to compare to monkey data we include only the three F1 values that had more than 10 recordings for each neuron. **B)** The computational scaffold of the RNN solution has 3 parts. The system encodes and stores the values of F1 in the blue region. Note, however, that the trajectories representing F1 = 20 and F1 = 40 fuse before the onset of F2. The network is still able to complete the task because all F2s that follow F1 = 20 and F1 = 40 are distinct. This means that the network uses only the information about F2 (orange region) to solve the task. For

the case of F2 = 15, however, the response of the system must vary depending on whether F1 = 10 or F1 = 20. Thus, the system must differentiate F1 = 10, and use that information to compare to F2 in the green region. **C)** Table of classification accuracies for both training and novel stimuli pairs. The computational scaffold in **B** predicts that the network will give erroneous results on the novel stimuli marked in red. The observed pattern of errors matches those predicted by LOOPER (top row), and confirms the assertion that the RNN fails to distinguish between F1 = 20 and F1 = 40 (as in **B**). A simulation of the LOOPER model fit to this network yields the same errors as the network (middle row). These error do not occur in an RNN trained on a modified dataset (Fig 3) in which the exploit cannot occur (bottom row).

scaffold, the RNN does not remember the distinction between F1 = 20 and F1 = 40, it will erroneously output "Greater than" when presented with F1 = 40 followed by F2 = 25. Similarly, the LOOPER model predicts that F1 = 20 followed by F2 = 30 should incorrectly yield the response "Less than" in the RNN. Finally, the computational scaffold extracted by LOOPER predicts that the RNN will give the correct response on the novel stimuli combinations F1 = 40 followed by F2 = 15, and F1 = 20 followed by F2 = 50. This is exactly what was observed for the RNN for these stimulus combinations on 99% of trials (Fig 2C). Simulation of the LOOPER computational scaffold (Methods) constructed from the RNN gave the same pattern of errors as observed in the original network. This shows that the LOOPER model accurately predicts responses of the network on new stimulus combinations. The failures of the RNN are not a reflection of noise in the network. If that were the case, then the outcome on the novel stimulus combination would be variable. In contrast, the LOOPER model predicts the specific stimulus combinations that lead to consistently wrong answers exactly, while other novel stimulus combinations result in correct performance. This suggests that the errors are due to the unexpected algorithm implemented in the RNN dynamics rather than stochastic influences or the generic tendency to overfit the training dataset by the RNN. Indeed, the RNN trained on a modified training set (discussed in Fig 3) outputs correct responses on all novel stimuli.

There are fundamental advantages to unsupervised modeling of the dynamics by LOOPER over supervised techniques. For instance, dPCA [13] of the RNN data shows that the trajectories reflecting F1 = 20 and F1 = 40 are statistically distinct during the interstimulus interval (Fig C in S1 Text). This would imply that the RNN does indeed encode and remember the F1 identity unequivocally. However, this dPCA-based assertion contradicts the specific pattern of errors made by the RNN on novel stimulus combinations. LOOPER, in contrast, correctly combines trajectories into a single trajectory. Even though statistical differences exist between the two trajectories, this difference was not large enough for LOOPER to find differences in the transition probabilities of the two trajectories. This reveals a fundamental distinction between statistical differences and the behaviorally salient aspects of neuronal dynamics. Statistical differences in neuronal activity do not in and of themselves imply that the system makes use of these distinctions.

## 3 LOOPER models generalize across distinct systems

Another significant advance of LOOPER is that it can, in principle, create a single model that generalizes across distinct dynamical systems performing similar computations. For this purpose, we will explore the dynamics that emerge in two distinct systems—an artificial RNN and primate prefrontal cortex—trained to perform a similar working memory task (Fig 2A). We apply LOOPER to previously published neuronal activity data from a working memory task in rhesus macaques [13, 35]. In this case, we modify the training set for the RNN such that each F1 shares a F2 value with at least one other F1. This keeps the RNN from using its "exploit" as in the previous section. LOOPER accurately reconstructs the monkey data with an $R^2$ of 0.97. The model can also predict future neuronal activity with a median $R^2$ of 0.9. The dynamics of the RNN were also well reconstructed ($R^2$ = 0.99) and accurately simulated ($R^2$ = 0.98) by

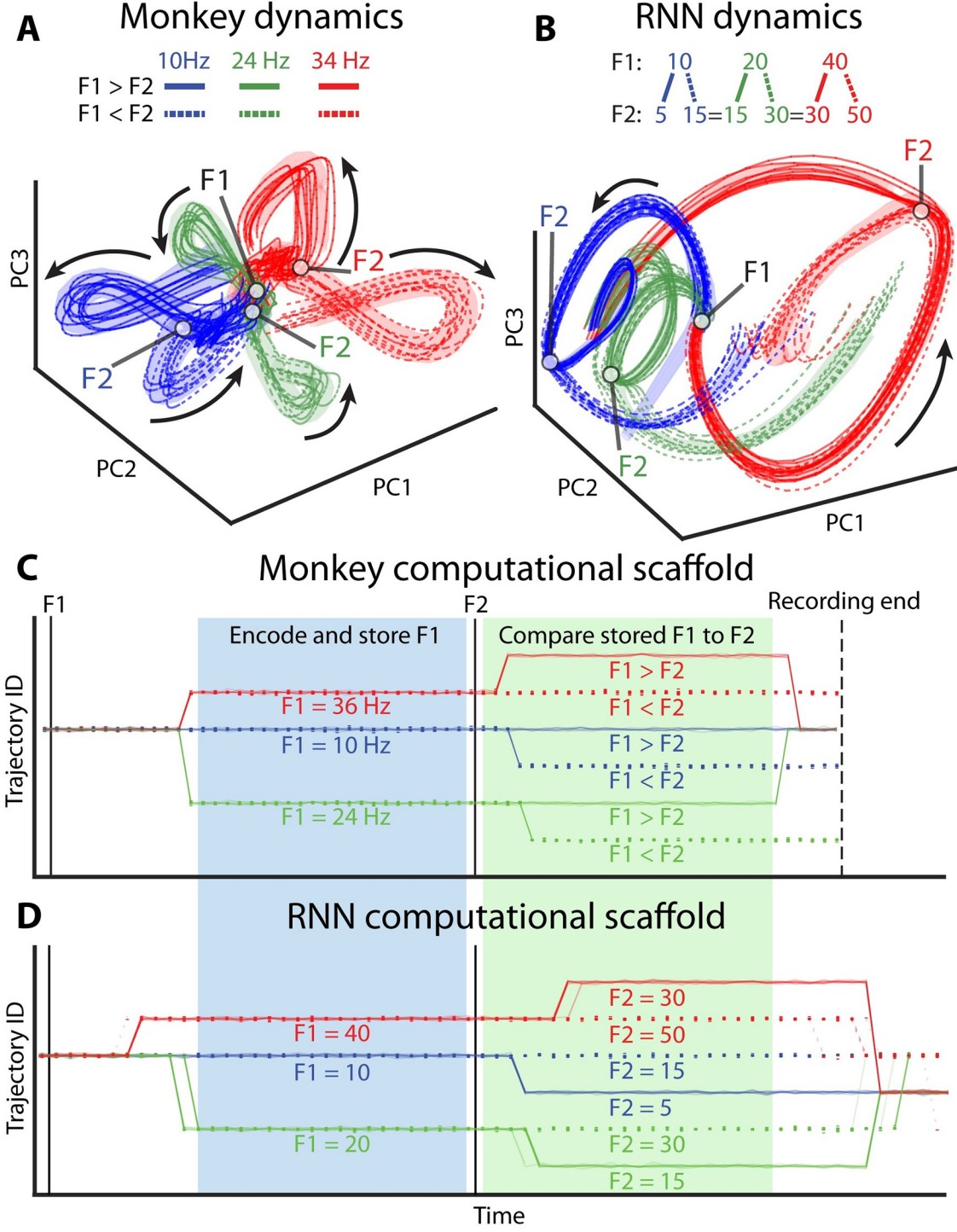

**Fig 3. Computational scaffold is conserved between monkey and RNN despite disparate neuronal activity patterns.** Mean subtracted activity in prefrontal cortex (**A**) and RNN (**B**) projected onto the first three principal components (56% variance explained in monkey, 83% explained in RNN). Thin lines show observed neuronal activity on a single trial (colored by task condition). Stable trajectories extracted by LOOPER are also projected onto these PCs(thick lines). Dashed lines indicate F1 < F2 and solid lines indicate F1 > F2. Shaded area reflects the variance of the data assigned to each model bin. The F1 and F2 stimulus timings are shown as white markers. Black arrows show phase velocity. Computational scaffolds constructed using LOOPER on monkey (**C**) and RNN (**D**) data. Both the RNN and monkey share the same number of trajectories and branching patterns, implying that they have the same computational scaffold. F1 causes the system to diverge into 3 distinct trajectories (Encode and store F1). F2 causes each of those trajectories to bifurcate (Compare stored F1 to F2). Note that the same F2 maps onto different trajectories depending on the value of F1. Thus, unlike F1 which is stored in the system dynamics, F2 information does not map onto a unique trajectory.

LOOPER. Quantitative accuracy of the LOOPER models and their predictive capabilities are discussed in detail in the final section.

A standard approach to quantifying similarities and differences in neuronal activity is to reduce the dimensionality of observations. Consistent with the fact that the underlying architecture and dynamics of individual neurons are fundamentally different between monkey brain and the RNN, projection of the data onto first three PCs reveals fundamental differences in the structure of neuronal activity that arises in these two systems trained on the same task (Fig 3A and 3B). To test whether this difference is limited to linear transforms of the data, we computed the redundancy index [36] of the canonical correlation analysis (CCA) which explicitly seeks to maximize the correlations between the monkey and RNN activity. The redundancy index for the monkey data was 25% variance explained ($\sim$2 PCs), and 51% variance explained (<1 PC) in the RNN data over all 175 canonical correlation dimensions. Thus, linear projections of the data that maximize the explained variance do not show similarities, while linear projections that maximize similarities between the data explain little of the variance. Differences in neuronal activity between the brain and the RNN are not entirely surprising. Architecturally distinct RNNs also produce distinct activity patterns even when trained on the same task [32].

The apparent differences in neuronal activity in the RNN and the brain may suggest that fundamentally different dynamics are engaged to solve the task in biological and artificial systems. Despite these apparent differences in neuronal activity, however, LOOPER reveals that the computational scaffold of both systems is identical (Fig 3C and 3D). In both systems, each unique combination of F1 and F2 cluster into distinct trajectories. Furthermore, trajectories that share F1 interlock for the duration of the interstimulus interval between F1 and F2. This trajectory structure maps in a one-to-one fashion to the distinct steps involved in the computation. The system starts off as a single point (start state). It then receives the first stimulus (F1) and quickly diverges into 3 distinct bundles corresponding to each F1 stimulus value (classifying F1, primate validation accuracy 99%, primate p-value < 0.0001 bootstrap vs. null model). The bundles remain distinct for the duration of the interstimulus delay (remembering F1, primate validation accuracy 98%, primate p-value < 0.0001 bootstrap vs null model). Upon receiving F2 each of these 3 distinct bundles further bifurcates into two—one branch representing F1 < F2 and the other representing F1 > F2 (comparing F2 to the remembered F1, primate validation accuracy 98%, primate p-value < 0.0001 bootstrap vs null model). Note that this bifurcation does not simply encode F2. The same F2 value yields distinct trajectories when encountered after different F1s. Thus, the dynamics depend not just on the stimulus, but also on the state of the system, which encodes the memory trace of F1. Finally, after the system produces a response, the 6 distinct trajectories representing each of the 6 conditions begin to converge (task end). This convergence reflects the fact that the information about F1 and F2 is no longer salient.

One advantage of LOOPER over other heuristic techniques is that the computational scaffold extracted by LOOPER is quantitatively accurate. This means that the timings of algorithmic events (splits and merges of the trajectories) are able to be approximated. The exact timings of these events would still require direct experimentation to confirm, as the LOOPER model assumes a level of stochasticity and makes use of state space reconstruction that both lead to the warping of temporal information.

The computational scaffolds in Fig 3D and 3E were constructed using a subset of data. We validate the scaffold by projecting the remaining subset of data not used in model construction onto the LOOPER model space (Methods, Fig D in S1 Text). The validation data matches the model's trajectory structure 96% of the time (n = 3960, p-value < 0.0001 bootstrap vs null model). Altogether observations in Fig 3A–3D imply that while the neuronal activity may be

distinct (Fig 3A and 3B), the differences between RNN and brain dynamics during working memory task are limited to the mapping from the same computational scaffold discovered by LOOPER to neuronal activity. In this way, LOOPER can be used to uncover essential similarities between seemingly distinct dynamics that emerge in structurally diverse neuronal networks.

When using the same stimulus statistics as those given to the monkey, the RNN learned a different solution to the task (Fig 2). Primates learn the more general algorithm, while the RNN solves the problem by exploiting stimulus statistics. This suggests that although the end result of RNNs and brains may be behaviorally similar on the training set, the learning rules that give rise to successful task performance, and therefore the structure of the computational scaffold, are fundamentally distinct. Direct comparisons between computational scaffolds revealed by LOOPER in distinct systems adapted to solve the same task may therefore help elucidate both the similarities and differences in how different networks learn and implement the computations necessary to solve the behavioral task.

## 4 LOOPER models produce falsifiable insights about the system

Because LOOPER naturally parcels neuronal dynamics into discrete trajectories, one useful application of LOOPER is the classification of behavioral responses based on the observed neuronal dynamics. To illustrate how this can be accomplished, we used BOLD signals recorded during a theory of mind task publicly available from the Human Connectome Project (Methods). Subjects are shown a 20s video that features shapes moving on the screen. In half the movies, the shapes move in such a way as to give the illusion that they are acting out a social drama [37], in the rest of the movies they move randomly and their interactions are strictly physical (e.g. bouncing off each other). It has been shown that the fMRI recordings can be used to decode whether the subject is performing the theory of mind task or other tasks from the human connectome project [38]. Here we attempt to identify neuronal processes that reflect the subject's beliefs about the presence of mental interaction. We focused on the most ambiguous movie in the dataset, where 66.2% of the participants thought that the shapes were not interacting, 10.9% of the participants thought that they were mentally interacting, and 25.2% of the participants were not sure. Choosing the most ambiguous dataset allows us to focus on the facets of activity that reflect subjects' beliefs about the stimulus rather than differences in the features of the video.

fMRI recordings were preprocessed (Methods) and bootstrapped across trials in which the subject answered "not interacting" versus trials in which they answered "interacting" before submitting to LOOPER. Note that LOOPER received no information about the decision made by the subject. Nonetheless, LOOPER predicts a divergence in the two trajectories ∼5s before the actual response occurs. These trajectories reliably segregate responses on the basis of the subject's belief concerning presence or absence of mental interactions in the movie (Fig 4B). Due to the low signal-to-noise ratio in the fMRI recordings, the exact timings of the trajectory divergence can be sensitive to parameter choices. Thus, we ran the LOOPER model over a range of parameters to test the robustness of the results. We find that separation of the two trajectories is least likely before the movie starts, is increased during the movie, and reaches its peak ∼5s before the subject response (p-values $< 0.001$). This pattern is consistent with the subjects accumulating evidence that supports their putative response (interaction vs no interaction) during the movie. To determine whether the LOOPER model is generalizable, we trained the model on 10 boostrapped trials from each condition. We then projected experimental observations from a non-overlapping validation dataset into the model space (Methods section **Validation of computational scaffold**) and attempted to decode the subject's beliefs about the stimulus from the trajectory assignment. This decoding is ∼95% accurate (Fig 4D).

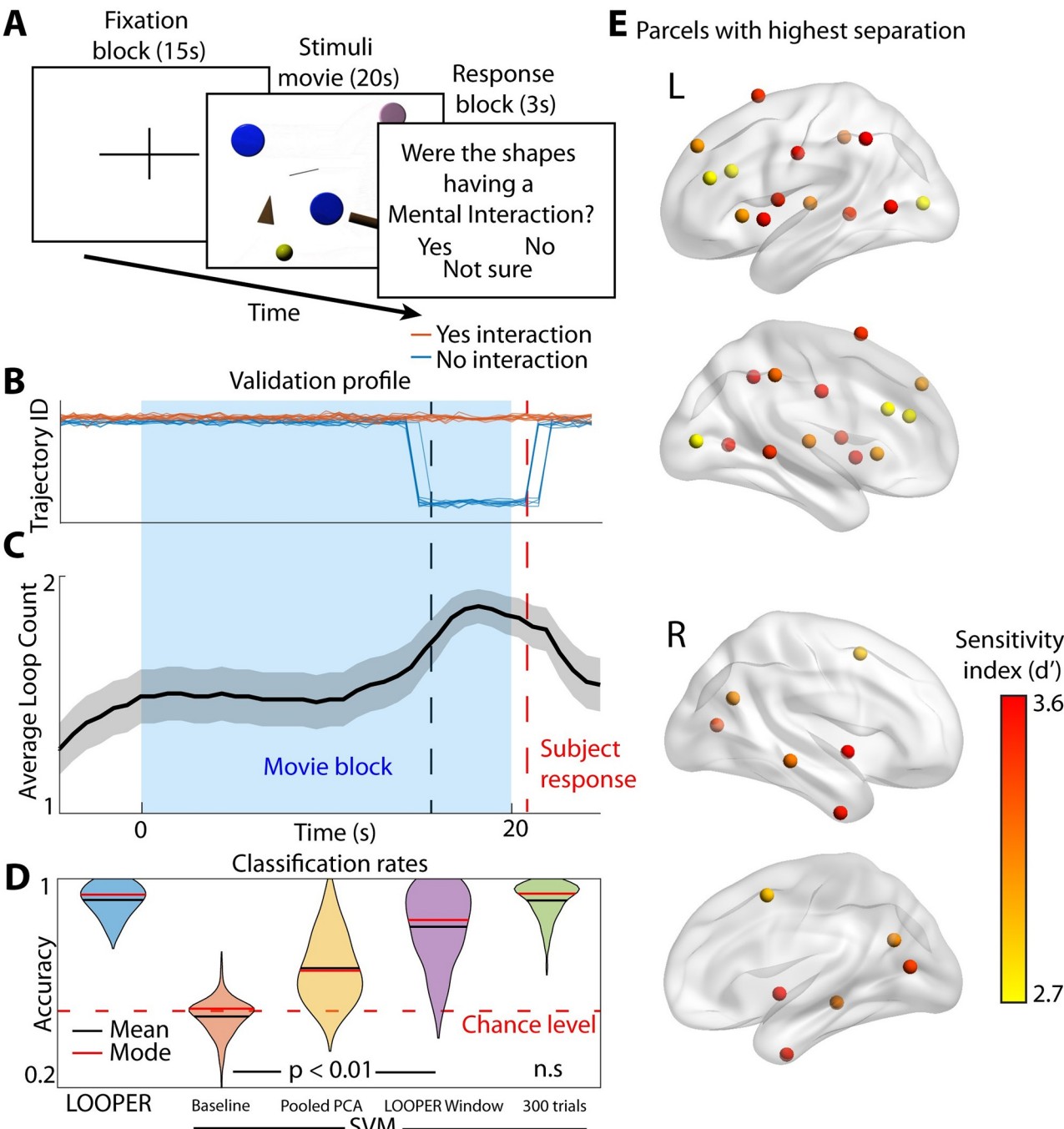

**Fig 4. Computational scaffold of neuronal dynamics extracted using LOOPER predicts choices on the theory of mind task. A)** Schematic of the "human theory of mind" task. Subjects are shown videos of shapes moving on the screen and are asked whether the shapes are having a mental interaction or not. We build the LOOPER model on only those trials in which the subjects answered either "Yes" or "No". **B)** Validation trials excluded from model construction are projected onto their closest model bin at each time step. A stereotypical example of the computational scaffold. Each trial in the validation set is colored by whether the subject responded that "Yes, they do think there was an interaction" (orange traces), or "No, they don't think there was an interaction" (blue traces). Notice that the two conditions diverge near the end of the movie and during the response window. This is consistent with accumulation of information throughout the movie that eventually culminates in a binary decision (validation accuracy 98.8%, n = 1000). **C)** Due to the high amount of noise in the fMRI signal the exact timings of divergence of trajectories is parameter dependent. To ensure that our result is robust with respect to parameter choices, we explore the effect of parameter values on the computational scaffold. The average number of trajectories and the 95% confidence interval is plotted at each time point. **D)** Accuracy of decoding the response using LOOPER and standard supervised techniques used in fMRI literature (SVM). All reported accuracies are on the validation dataset. The baseline SVM classifier used the same training data as LOOPER (10 trials of each condition bootstrapped over 60 subjects). For LOOPER, the raw parcel time series were used. For SVM, the

data were subjected to PCA (top 20 PCs, 98% variance explained) and averaged over time. The baseline model (orange) averages over the full trial time. Classification accuracy is vastly improved by using the top 20 PCs calculated on the basis of both the training and validation datasets (yellow). Accuracy is further improved by taking the average over the period of separation found by LOOPER (**B** and **C**) instead of over the full trial (purple). However, none of these models perform at the same level as the LOOPER model (p < 0.001). We can recover LOOPER-level accuracies by dramatically increasing the number of trials fed into the SVM (green). **E)** Visualization of the most sensitive parcels during the time of divergence of the two conditions (responded "Yes" or "No"). We generate 100 random sets of trials using the same bootstrapping paradigm as used to build the LOOPER model. For each set of trials we find the maximum sensitivity index, d', for each parcel during the time of interest and take the median of these maximum d' values over the 100 iterations of trials. The top 20 brain regions are plotted.

We then compared the decoding accuracy attained by LOOPER to conventional support vector machines (SVMs) routinely used in analysis of fMRI datasets. We applied the same preprocessing to the data as used in LOOPER. We first fit the SVM to the full time series of the preprocessed fMRI data, but the model was overfit (near perfect accuracy on the training data, chance level on the validation data). The standard fMRI analysis technique of averaging activity over the full trial also failed to solve the overfitting problem. Thus, to reduce the degrees of freedom we projected the preprocessed fMRI data onto principal components analysis (PCA) space before averaging (20 components, 71% variance). Note that LOOPER did not require this preprocessing step. Preprocessing the data with PCA did not improve classification performance significantly (Fig 4D, orange). However, when the data from the validation dataset was added to the calculation of PCA, the classification performance of the SVM model improved dramatically (Fig 4D yellow, p-value <0.001). Again, LOOPER was trained on just the raw parcel data. LOOPER predicts that the separation of trajectories only occurs during the second half of the trial. Thus, we might expect increased classification accuracy during this time period. To test this hypothesis, we fit the SVM model using PC weights averaged over the time period of trajectory separation (Fig 4B and 4C, dashed lines). Indeed, this dramatically increased performance of the SVM classifier (Fig 4D, purple). This increased classification accuracy provides independent validation that LOOPER correctly identifies the time window during which consistent differences in BOLD signal emerge on different task conditions.

None of the SVM models discussed thus far reach LOOPER's level of classification accuracy on the validation set (p-value < 0.001). The discrepancy between the near perfect training accuracy and the reduced validation accuracy suggests that SVMs are over fitting the data. The HCP includes 1029 participants. After excluding participants with missing data and those that answered "Not Sure", we were left with 771 participants. Only 109 of these participants answered "No". Thus, while increasing the size of the dataset is likely to improve the performance of the SVM classifier, obtaining a dataset large enough to make application of these standard techniques robust would be impractical. To estimate the amount of data required to recover the accuracy offered by LOOPER we increased the number of bootstrapped trials fed into the SVM classifier to 600 (300 per condition) (Fig 4D, green). At this level of data the supervised classifier does produce comparable results to LOOPER—which achieved its accuracy using only 20 trials (10 per condition). Note that the HCP dataset is one of the largest fMRI dataset to date, and that the number of trials used to match LOOPER performance exceeds the total number of trials available in the HCP dataset.

Since the LOOPER model can be readily mapped back to neuronal activity, we were able to identify the brain regions that exhibit reliable differences in activation depending on whether the subject believed that the movie contained mental interactions or not. This was accomplished by quantifying the differences in neuronal activity between the two trajectories of the LOOPER model as maximum $d'$ observed during the period where trajectories are reliably separated (Fig 4B, black dashed line to red dashed line). To ensure robustness, we take the

median of this maximum value computed over 100 bootstrap trials. The 20 brain regions with the highest $d'$ are shown in Fig 4E. Note the striking asymmetry between the cerebral hemispheres. Brain regions that most reliably distinguished between trials in which the subject believed that there were mental interactions were predominantly located in the left cerebral hemisphere. Furthermore, regions located in the vicinity of areas associated with language processing [39] and social cognition (including the dorsomedial prefrontal cortex and temporal parietal junction) [40, 41] exhibited differential activation. This result is consistent with the notion that the subjects were creating a narrative account of the mental interactions.

## 5 LOOPER models accurately reconstruct and predict neuronal activity

LOOPER achieves its greatly improved interpretability and generalizability by assuming that the data can be well approximated by a collection of one dimensional trajectories. To determine whether this assumption significantly degrades the ability to reconstruct complex neuronal activity from a simple LOOPER model, we constructed models of several artificial and experimental neuronal systems: noisy RNN trained on a working memory task (Fig 5A), pan-neuronal calcium imaging in *C. elegans* (Fig 5B), visual evoked local field potentials recorded using high-density electrode arrays in mouse (Fig 5C), firing of neurons in the prefrontal cortex of a primate during working memory task (Fig 5D), and blood-oxygen-level-dependent (BOLD) signals in human fMRI experiments during a theory of mind task. Details of the datasets and tasks used can be found in the Methods section.

To quantify the ability of LOOPER to reconstruct neuronal activity, we first mapped neuronal activity onto the LOOPER model and then mapped from the LOOPER model back to neuronal activity (Methods section **Validation of computational scaffold**). Note that while preprocessing steps make use of PCA to reduce the spurious noise in the time series data, all predictions and reconstructions are on the level of individual channels (units in RNNs, neurons in *C. elegans* and primates, electrode in mice and parcels in humans). The reconstructed time series were remarkably similar to the experimental observations and have correlations at or above $\sim 0.8$ for all datasets (computed as correlation across all signals). Thus, condensing the neuronal dynamics onto a simple LOOPER computational scaffold does not result in an appreciable loss of information about the neuronal activity. We further demonstrate that this approach is applicable to a wide variety of biological and artificial systems, as well as recordings performed at vastly different spatial and temporal scales, from single neurons to fMRI BOLD signals.

LOOPER is not only a dimensionality reduction tool, but a model capable of successfully predicting future experimental observations. To determine the accuracy of these predictions, we split the datasets in Fig 5A–5E into equal non-overlapping subsets for training and validation. Each dataset had samples from each condition. LOOPER models were constructed on the training subset. The model was then validated by its ability to predict the dynamics of a bootstrapped sample of trials from the validation dataset (A full list of parameters used in preprocessing, along with the LOOPER parameters for each dataset can be found in Tables 1 and 2). To initialize the simulation, a point in the validation sample was mapped to the closest model bin (beginning of a solid black line in Fig 5F–5I). The LOOPER model was then simulated in the model space starting from these initial conditions and mapped back to neuronal activity (solid red line). Because LOOPER is a stochastic model, mean values and variance ($\sigma$, red shaded areas) can be estimated from multiple simulations, starting from the same initial conditions. LOOPER simultaneously simulates activity in all recording channels (4 sample channels are shown in Fig 5F–5I). The distribution of correlations computed across all recording channels and all task conditions is shown in the bottom row of Fig 5F–5I. Note that the trials used

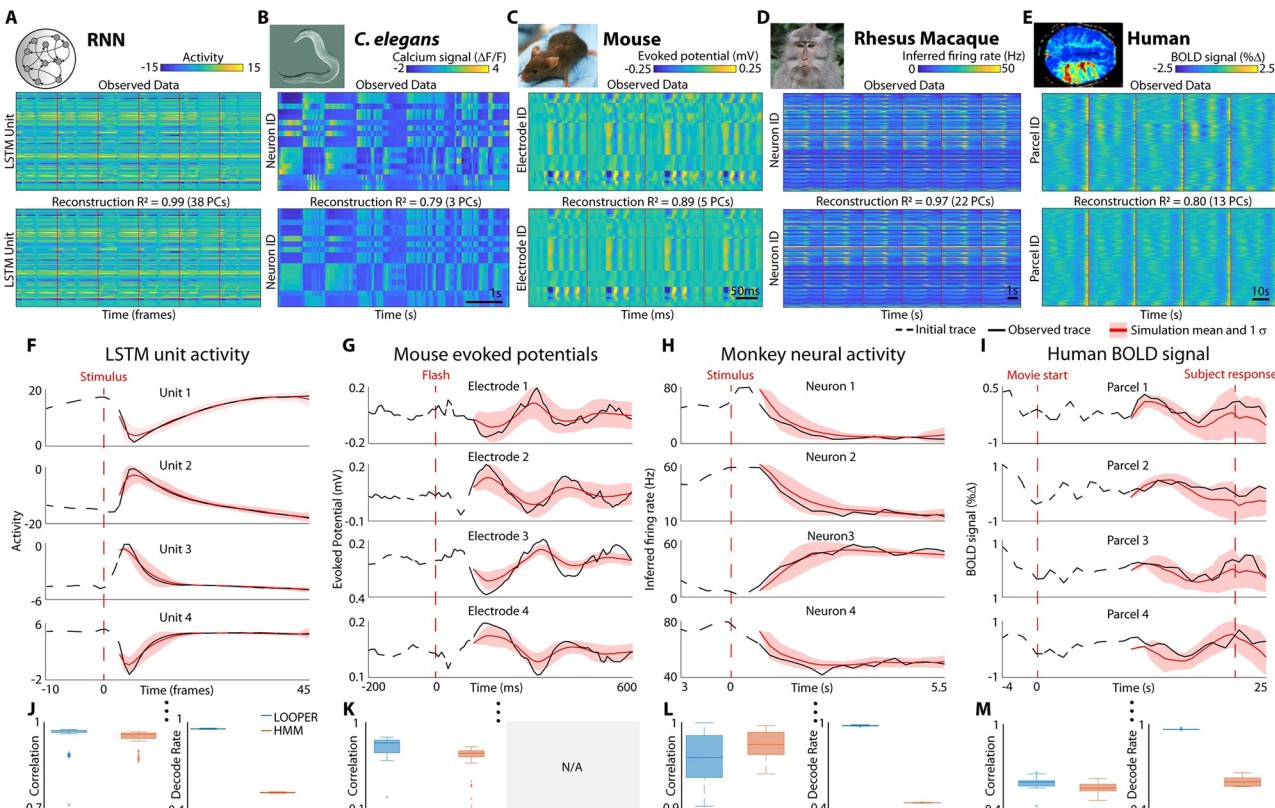

**Fig 5. LOOPER model explains majority of variance and predicts future neuronal activity.** Simple models obtained using LOOPER are remarkably good at reconstructing the dynamics observed in diverse systems such as artificial recurrent neural networks (**A**), whole brain calcium imaging in *C. elegans* during spontaneous locomotion (**B**), visually evoked local field potentials in the mouse (**C**), firing of neurons in prefrontal cortex of a primate during working memory task (**D**), and BOLD signals in humans performing a "theory of mind task" (**E**). Top panels in (**A**-**E**) are experimental observations (red vertical lines show different trials or stimuli timings). Bottom panels are LOOPER reconstruction of the observed activity. $R^2$ values are calculated between each observed trial (i.e. #channels x time matrix) and the corresponding reconstructed trial. We also include the number of PC components required to achieve an equivalent $R^2$ in the parentheses. LOOPER models preserve the majority of variance in the data across a vast variety of systems and signal types. LOOPER is also able to predict future neuronal activity of the systems (**F**-**I**). Black lines are either a single trial (RNN) or a bootstrapped sample of several trials (mouse → 5 trials; monkey → 10 pseudo trials (see main text); BOLD signals → 60 trials). Top four panels show examples of four recording channels (total of 100 LSTM units, 49 electrodes in mouse, 179 neurons in monkey and 264 parcels in human). Red lines show an average of 100 simulations of the LOOPER model (shaded area shows standard deviation). The bottom panels show the distribution of correlation coefficients computed across multiple bootstrap subsets (or individual trials) and recording channels. For each experimental system, the LOOPER model is constructed on a subset of trials. Validation trials (or bootstrapped averages) are constructed from a non-overlapping subset of trials. Each simulation begins at the same timepoint in the trial (beginning of solid black line). The initial conditions for the LOOPER simulation are given by the closest model bin to the observed neuronal activity at this time. The dynamics starting from these initial conditions given by the LOOPER model are simulated and projected back into observation space for comparison to experimental observations. Note that the simulation periods are chosen such that no task-relevant stimuli are present, and so no input is required for the simulations. Correlation is computed over the period of the simulation starting from $t_0$ and continuing for several time steps (RNN → 40 time steps, mouse → 50 time steps, monkey → 19 time steps, human → 20 time steps). Median correlation values are RNN → 0.98, mouse → 0.79, monkey → 0.95, human → 0.62, (**J-M**, blue box, left). We also compared LOOPER's simulations to simulations using conventional HMMs using the same methodolgy as above (**J-M**, orange box, left). Finally, we used both the LOOPER and HMM models to attempt to decode the state of the system at each point in time and task condition. The task decoding rate is the average difference between the model's decoding rate and the expected decoding rate given the task (Romo task: 50% during the interstimulus period and 100% after the presentation of F2, Theory of Mind task: 100% during the period of separation in Fig 4B). Average decoding rates are shown in Fig 5J–5M (blue and orange boxes, right). Chance level decoding rate given a uniform distribution null model of the Romo task is 25%, and for the Theory of Mind task it is 50%. Note that the mouse data has no associated task and so is left blank.

in this figure are bootstrapped subsets of trials. Machine learning approaches [14] have recently succeeded in accurately modeling single trials of neuronal activity. The LOOPER model used to make the predictions in Fig 5 can also accurately simulate neuronal activity observed in a single trial during a working memory task in primate (mean correlation = 0.775,

**Table 1. Parameters used for preprocesssing.**

| System | # PCs | Bootstraps | Conditions | Trials | Use z-score | Smoothing | Embed delays | Embed count |
|---|---|---|---|---|---|---|---|---|
| RNN | 20 | 0 | 6 | 10 | F | 1 | 2 | 10 |
| *C. elegans* | 0 | 0 | N/A | Contiguous | T | 1 | 10 | 10 |
| Mouse | 8 | 5 | 1 | 50 | F | 1 | 5 | 4 |
| Primate | 10 | 10 | 6 | 10 | F | 2 | 2 | 11 |
| Human | 0 | 60 | 2 | 10 | T | 4 | 2 | 5 |
| RNN (Modified) | 0 | 0 | 6 | 10 | F | 2 | 2 | 5 |
| RNN (Noiseless) | 0 | 0 | 6 | 10 | F | 1 | 0 | 0 |

Preprocessing applied to experimental datasets before submission to LOOPER.

Fig E in S1 Text). Thus, simple LOOPER models can successfully predict complex neuronal signals recorded at different scales and across distinct biological and artificial systems.

LOOPER falls into a general modeling methodology called the Hidden Markov Models (HMMs), but differs significantly from conventional HMM implementations in the way the transition probability matrix is constructed and then simplified. Standard HMM algorithms find the set of states and transition probability matrix under which the observed data is most probable. In contrast, LOOPER follows a topological data analysis approach to build up a model of the dynamics based on local transition probabilities and the assumption that trajectories are stable. To examine how the modifications used by LOOPER affect both quantitative performance and interpretability of the model, we compared LOOPER to conventional HMM trained using the Expectation-maximization [42] algorithm implemented in the Bayes Net Toolbox [43]. Each dataset was preprocessed using the same state space discovery method as used for the LOOPER model and after applying standard symmetric diffusion mapping (S1 Text). Bayesian optimization was used to identify optimal hyper-parameters. We find that LOOPER and HMM are both able to reconstruct and model the data with comparable accuracy (Fig 5J–5M, left). However, LOOPER drastically outperforms HMM on decoding the task condition from the model state (Fig 5J–5M, right). Thus, the higher interpretability of LOOPER models does not come at the expense of high quantitative accuracy and the ability to predict future neuronal activity.

**Table 2. Parameters used in LOOPER models.**

| System | Diffusion mapping | | | | Model reduction | | Find loops | |
|---|---|---|---|---|---|---|---|---|
| | Neighbor count | Use local dims. | Repop. density | Min. return time | Distance measure | Max. check time | Total state count | Use terminal state |
| RNN (Original) | 7 | T | 0.95 | 10 | corr. | 5 | 300 | T |
| *C. elegans* | 8 | T | 0.95 | 10 | corr. | 10 | 25 | F |
| Mouse | 20 | T | 0.5 | 10 | corr. | 10 | 200 | T |
| Primate | 7 | T | 0.95 | 10 | corr. | 10 | 200 | T |
| Human | 4 | T | 0.95 | 10 | corr. | 5 | 100 | T |
| RNN (Modified) | 7 | T | 0.95 | 10 | corr. | 10 | 300 | T |
| RNN (Noiseless) | 10 | T | 0 | 10 | corr. | 10 | 100 | T |

For cluster and loop count parameters (not listed in table) we use a range of values such that a minimum MDL value is observed in the middle of the range. Boolean values are listed as T for true and F for false. Note that the "Use local dimensions" and "Distance measure" parameters are provided for convience only, and were not modified on any of the above data sets.

## Discussion

Here we described a methodology that extracts simple models of dynamics that emerge in neuronal networks adapted to perform a behavioral task. The model accurately reconstructs and predicts neuronal activity acquired from a variety of complex biological and artificial neuronal networks and scales of observations from single neurons in an immobilized nematode performing fictive locomotion behaviors to fMRI in humans performing a theory of mind task. The structure of the model is closely related to the information that is expressed in the network dynamics and how this information is processed. One fundamental advance of our methodology is that the same LOOPER model can apply to distinct neuronal networks. The ability to generalize a single model across multiple distinct neuronal networks is likely due to the close relationship between the structure of one dimensional trajectories extracted by LOOPER and the computations performed by neuronal networks in service of the task.

This work improves upon previous results in *C. elegans* [6] in which loops were defined using the most stable complex eigenvalue of the diffusion map. A similar approach has also been applied to find rotations in PCA space [8]. While successful in some systems, these spectral methods are not likely to generalize because eigenvalues are associated with a single phase velocity and cannot easily model dynamics that unfold on multiple time scales. LOOPER solves this problem by explicitly finding trajectories, regardless of the time scale.

The study of brain dynamics is complicated by the sheer number of parameters needed to adequately capture the complexity of biological systems and their nonlinear interactions. Thus, detailed biophysically realistic models are experimentally intractable and conceptually unrevealing in most systems of interest. One successful strategy used to gain insight into the relationship between brain dynamics and computations enacted by them is to replace one complex nonlinear dynamical system—the brain—by a more tractable dynamical system such as RNNs. Machine learning techniques are then used to train the RNN on a similar behavioral task [5, 44]. Thus, rather than studying the experimentally observed dynamics directly, one can study the dynamics of the RNN under the assumption that similar task demands lead to the emergence of similar dynamics. There are many notable advantages of the machine learning approach, such as emergence of more realistic complex neuronal activity patterns that resemble those experimentally observed in the brain [45]. The main advantage of the machine learning approach, however, is that the dynamics of artificial RNNs can be better understood through stability analysis. The studies of RNN dynamics revealed a close link between the computations enacted by them and the structure formed by the attractors and the transients that connect them.

While the study of the relationship between dynamics and computations in artificial RNNs is in its own right an exciting field, extrapolating the conclusions from the analysis of the RNNs to the brain relies on a key assumption that dynamics of the RNNs will be essentially similar to those observed in the brain so long as both systems are performing the same task. This may be true in relatively simple tasks that have been previously analyzed using this approach. However, as we show here, even in simple tasks, subtle differences in training can result in distinct computational scaffolds. In more complex tasks, such as spatial navigation, RNNs can learn distinct dynamics depending on the specifics of the pre-training algorithm [46]. In general, it is not guaranteed that dynamics that are learned in biological and artificial systems will exhibit the same computational scaffolds.

While the kinds of phenomenological modeling implemented by LOOPER does not directly allow us to estimate stability of the system in a detailed way, the features uncovered by LOOPER are closely related to those discovered using stability analysis in RNNs (Fig B in S1 Text). This is not entirely surprising because LOOPER explicitly combines states of the system

into neighborhoods if the trajectories initiated from these states tangle under the influence of noise. This tangling will occur if two different locations in state space are being attracted by the same stable fixed point or, alternatively, repelled by a saddle point in the same direction [47, 48]. In contrast to the formal stability analysis that can only be applied to model systems, LOOPER extracts a set of stable trajectories and identifies the branching points solely on the basis of experimental observations. The ability to extract simple models in both experimental and model systems can allow for the direct comparison between the kinds of computations that are learned in biological and artificial systems as well as in distinct biological networks.

LOOPER builds upon elements of several existing modeling techniques to construct the computational scaffold. For instance, the manifold inference from neuronal dynamics (MIND) [3] algorithm also uses diffusion maps to extract a low dimensional manifold from high dimensional neuronal activity. While MIND attempts to relate the position of the system in the manifold space to behaviorally salient variables and neuronal activity, LOOPER specifically focuses on identifying trajectories that unfold in the manifold space. As we show here, each trajectory maps clearly onto a single salient task condition. In spirit, the motivation behind LOOPER is similar to techniques such as switching linear dynamical systems and adaptive locally linear models [34, 49]. Both LOOPER and these methods attempt to express the overall system dynamics as a collection of simpler systems. There is an important distinction, however, in how LOOPER partitions the overall dynamics into simpler subsystems. Both adaptive locally linear models and switching linear dynamical systems approximate the overall behavior as a collection of linear dynamical systems. In contrast, LOOPER specifically seeks to approximate the system as a collection of one dimensional trajectories regardless of whether the dynamics along them are linear. This approximation is motivated by the fact that noise degrades information stored in nonlinear dynamical systems and on empirical results suggesting that neuronal trajectories specifically avoid divergence and tangling [9, 23]. As we demonstrate, this approximation works well for a number of experimental settings. However, the specific goals of the model depend on its application. For example, in brain-machine interface, quantitatively accurate prediction and decoding of neuronal signal, rather than interpretability, is the primary goal. While LOOPER is a quantitatively accurate model in its own right, given sufficiently large data sets, machine learning approaches such as LFADS [14], HMMs [50, 51] and SVM [52] are likely to offer superior quantitative accuracy. In contrast to machine learning methods, LOOPER offers significant advantages in CTD applications, as the models constructed by LOOPER are interpretable. Another advantage of LOOPER over machine learning techniques is that the models can be built on datasets with an experimentally tractable number of observations.

The fundamental assumption behind LOOPER is that neuronal dynamics can be well approximated by a set of one dimensional trajectories. In simple experimental paradigms that require classification of sequences of inputs, this assumption works well in a variety of systems (including the Lorenz system, Fig F in S1 Text). It is less clear whether the one dimensional approximation will also suffice for more complex tasks that involve open-ended recurrent interactions with the environment. For instance, in spatial navigation, it is possible that the approach exemplified in MIND will offer more insight into the computations carried out by neuronal dynamics. Future work will examine whether the one-dimensional assumption behind LOOPER breaks down on complex open-ended tasks. Many real world behaviors, however, involve choosing among distinct alternatives based on a sequence of stimuli [35, 48]. Furthermore, such tasks are routinely used in the laboratory settings to probe various aspects of cognitive processes implemented in neuronal dynamics [53–55]. Lastly, we made several methodological choices in attempting to identify one dimensional trajectories in neuronal recordings. As this is a highly non-trivial problem for

high dimensional, incomplete, and noisy neuronal recording data, it is likely that future work will improve on the specific methodologies used to uncover one dimensional trajectories. Regardless of the specific methodological choices, the finding that neuronal dynamics across distinct systems engaged in a broad class of behaviorally salient tasks can be approximated by simple interpretable models is an important advance in linking neuronal dynamics to computation.

## Methods

### 1 Ethics statement

Mouse Data: All experiments in this study were approved by Institutional Animal Care and Use Committee at the University of Pennsylvania and were conducted in accordance with the National Institutes of Health guidelines.

fMRI Data: See Open HCP Dataset for ethics statement.

Monkey Data: See Romo et al. [35] for ethics statement.

*C. elegans* Data: See Kato et al. [56] for ethics statement.

### 2 Overview of LOOPER

A summary of LOOPER can be found here, and in Fig G in S1 Text. A rigorous mathematical description is given in the S1 Text. LOOPER consists of the following three steps. The first step involves the construction of the diffusion map (S1 Text section **Asymmetric diffusion mapping**). This diffusion map is then subjected to two clustering steps. In the first clustering step, individual states of the diffusion map are hierarchically clustered to arrive at a compressed map (S1 Text section **Reducing the matrix**). The number of clusters is optimized using a standard minimal description length algorithm. In the final step, which implements the one dimensional approximation, repeated sequences of states are clustered into trajectories on the basis of a modified edit distance (S1 Text section **Loop Finding**). The number of trajectories is determined by optimizing the data reconstruction accuracy using an estimate of the maximum likelihood.

The method requires three parameters: size of the local neighborhood, repopulation density of the diffusion map, and the total number of bins of the final model. The first two parameters determine how the diffusion map is constructed, while the last one sets how the continuous dynamics along the trajectories are approximated. In Fig H in S1 Text, we explore the effect of different parameter choices on the reconstruction accuracy of the model. Only the size of the local neighborhood has an appreciable effect on model performance. The size of the local neighborhood is set by the number of nearest neighbors that go into the calculation of the diffusion map kernel (S1 Text section **Partitioning the data into local neighborhoods**). Thus, this parameter is closely related to the number of similar trials observed in the data set. So long as the number of nearest neighbors is approximately equal to the number of trials, the behavior of the model is robust.

LOOPER assumes that the observations of neuronal activity are sufficient to recover the true state space of the system. While in some systems the observed neuronal activity is sufficient to approximate the state space, this is not always the case. In the S1 Text section **State space extraction** we describe how state space can be nonetheless approximated given incomplete observations of neuronal activity.

The first step of LOOPER is to reduce the complexity of the nonlinear dynamics by finding a representation over which the dynamics evolve in a linear fashion. This is accomplished using diffusion maps. Transition probabilities between observed data points in a local

neighborhood are proportional to the rate of diffusion.

$$P(\mathbf{x}_1, \mathbf{x}_2) \sim exp\left(-\frac{D(\mathbf{x}_1, \mathbf{x}_2)^2}{2\sigma^2}\right), \tag{3}$$

where $D$ is a distance measure and $\sigma$ is a normalization term that sets the size of the local neighborhood. We estimate the size of each local neighborhood using the parameter: Number of Nearest Neighbors, $nn$. See sections **Distance measure** and **Partitioning the data into local neighborhoods** in the S1 Text for specific details on how we calculate $D$ and $\sigma(nn)$ respectively. This approach is suitable for purely diffusive systems where there is no net flow. In order to adapt diffusion maps to ordered time series, we center the diffusion kernels on the next observed time step, $\mathbf{x}_{t+1}$. The diffusion probability computed in this fashion for all experimental observations forms a transition probability matrix.

The next step in the LOOPER algorithm is to coarse grain the state space $\mathbf{X}$ using hierarchical clustering of the states of the diffusion map. The similarity between states $\mathbf{x}_i$ and $\mathbf{x}_j$ is defined as follows:

$$S(\mathbf{x}_i, \mathbf{x}_j) = \rho(P(\mathbf{x}_{n\in\mathbf{X}}|\mathbf{x}_i), P(\mathbf{x}_{n\in\mathbf{X}}|\mathbf{x}_j)) \tag{4}$$

where $\rho$ is the correlation coefficient and $P(\mathbf{x}_{n\in\mathbf{X}}|\mathbf{x}_i)$ is the *i-th* row of the diffusion map which encodes a distribution of states expected after one time step starting from $\mathbf{x}_i$. Thus, if the probability distribution of states found one time step starting from two distinct states is similar, then information about the difference in initial conditions is quickly forgotten and the two states can be agglomerated into a single cluster. In this way, the distance measure is conceptually similar to the definition of tangling in Russo et al [9].

The number of clusters in this coarse grained diffusion map is optimized using the objective function $I_{\text{loss}}$, which quantifies the information lost from the original diffusion map. Each row of the original and the coarse grained diffusion map, after appropriate normalization, is a probability distribution function. To compare these distribution functions between the original and the coarse grained maps, we use a standard information-theoretic measure, Kullback-Leibler divergence (KL divergence). For computational efficiency, we only consider the upper bound of this measure (S1 Text section **Reducing the Matrix**). Diffusion maps describe the evolution of the system after one time step. Thus, comparison of the original and coarse grained maps directly reveal similarities in short term behaviors, but obscures differences in longer term behaviors. This is appropriate for measuring tangling but not divergence [23]. To assure that the coarse grained and the original diffusion maps have the same patterns of divergence, we minimize $I_{\text{loss}}$ over a range of time scales. The final number of clusters in the coarse grained map is chosen such that $I_{\text{loss}}$, penalized by the number of parameters, is minimized. The penalty is implemented using a standard minimum description length algorithm (S1 Text section **Reducing the matrix**).

$$\text{MDL} = I_{\text{loss}} + \frac{k}{2}\log\frac{n}{2\pi}, \tag{5}$$

where $k$ is the size of reduced model and $n$ is the size of original transition probability matrix. The result is now a reduced representation of the dynamics of the system that still explains as many features of the original transition probability matrix as possible. This coarse grained matrix is explicitly constructed to minimize tangling and preserve the divergence observed in the original observations.

A natural consequence of minimizing tangling and divergence is that trajectories formed by the system will be approximately one dimensional objects. Thus, the last step of

the algorithm identifies repeated sequences of states emitted by the coarse grained diffusion map and combines them into trajectories. In general clustering trajectories is difficult because it is not clear how to compare trajectories that start and end in different states. We circumvent this issue by focusing instead on "loops" formed by trajectories. Loops can be readily defined by the most likely path from each state back to itself in the coarse grained matrix. Thus, loops are sequences of states that start and end in the same state. In contrast, trajectories are sequences of states that may have different start and end states. Trajectories can be readily inferred from loops by noting that when two loops share the same clusters they are indistinguishable. If two loops separate and then merge back together they produce two trajectories—one that is a full loop, and one that exists only when they two loops deviate (red trajectory, Fig 1E). We will use the terms "trajectory" and "loop" interchangeably in the description of the method.

To perform this clustering we construct an edit distance from state sequences emitted by the coarse grained model,

$$S_{\text{loop}}(i,j) = \prod_{c_k \in W_i} \max_{c_l \in W_j} S_{\text{cluster}}(c_k, c_l), \tag{6}$$

where $S_{\text{loop}}(i,j)$ is the similarity between two state sequences, $W_i$ is a set of states comprising the $i$-th sequence, and $S_{\text{cluster}}$ is the similarity between two states. See section **Finding loops** for details on finding trajectories and the definition of $S_{\text{cluster}}$.

The above similarity measure is used to perform hierarchical clustering. To identify the number of trajectories in the final model, we optimize the fit over a range of putative trajectory counts. To accomplish this, dynamics along each trajectory, $\alpha$, are binned along phase, $\theta$. Each one of these bins contains a set of experimental observations. To determine how much information is lost by approximating the neuronal dynamics as a model with a certain trajectory count, we first assign each experimental observation to the most likely model bin. We then emit the mean of all observations falling into a particular model bin. The goodness of fit is quantified as the estimated log likelihood of the data given the model across all bins (See S1 Text section **Assembling Loops into the complete model**). The number of trajectories that maximizes goodness of fit is chosen for the final model.

The net result of this process is a simple linear model $(\alpha, \theta)_{t+1} = \mathbf{M} \cdot (\alpha, \theta)_t$, where $\mathbf{M}$ is a Markov matrix. Once a system starts out in a particular trajectory, $\alpha$, it evolves in an ordered fashion along $\theta$ until a point where multiple $\alpha$'s join. At this point the system switches, withsome probability, from one $\alpha$ to another. Thus, the net result of the LOOPER algorithm is a simple description of the dynamics that consists of a system of interlocking stable trajectories.

## 3 Neuronal data

Primate data was processed using the same pipeline as in Kobak et al. [13]. *C. elegans* data makes use of data published by Kato et al. [56], and follows the same pipeline as in Brennan 2019 [6]. Mouse data was collected by electrophysiological recordings in the Proekt Lab. Signals were filtered between 0.1 and 325 Hz, and down-sampled to 1000 Hz. Noise channels were removed, and the signal was referenced to the mean signal. fMRI data was obtained from the Open HCP dataset with permission. Preprocessing was done using the HCP MRI data preprocessing pipelines [57]. The brain regions of interest (ROIs) were defined by Power et al. [58]. More details can be found in the S1 Text sections **Preprocessing of primate data**, *C. elegans* data, **Mouse data**, and **fMRI data**.

**Table 3. Hidden Markov model hyperparameters.**

| System | Covariance | # PCs | Epsilon | t | k | Total States |
|---|---|---|---|---|---|---|
| | | | Optimized parameters | | | |
| RNN | 0.2 | 49 | 14 | 22 | 14 | 66 |
| Mouse | 0.1 | 31 | N/A | N/A | N/A | 150 |
| Monkey | 0.1 | 49 | 35 | 18 | 10 | 44 |
| Human | 0.2 | 4 | 23 | 2 | 14 | 83 |

Parameters used to construct the Hidden Markov models used to compare to reconstruction accuracy attained with LOOPER. # PCs, Epsilon, t, k and Total States were all optimized using Batesian Optimization. Monkey data achieved superior results when diffusion mapping was not applied to the data.

## 4 Validation of computational scaffold

The validation data is projected onto the LOOPER computational scaffold by finding the closest model state $(\alpha, \theta)$ for each observed time point. For each time point, the locally z-scored distance to each cluster is found:

$$D_{local}(t, i) = \| (\mathbf{x}_t - \bar{\mathbf{x}}_i)/\sigma_i \|_2^2 . \tag{7}$$

Where $t$ is the time of the observed data point, $i$ is the index of the model bin, $\mathbf{x}_t$ is an experimental observation at time $t$, $\bar{\mathbf{x}}_i$ and $\sigma_i$ are the mean and standard deviation of all points in $i$-th model bin respectively, and $\| \cdot \|_2^2$ is Euclidean distance. To assign $x_t$ to $\bar{\mathbf{x}}_i$, the minimum of $D_{local}$ over model bins is found. Each time point can then be unequivocally assigned to the phase and the trajectory identity $\bar{\mathbf{x}}_i \mapsto \{\theta, \alpha\}_i$. Once each time point is assigned a trajectory ID the computational scaffold can be plotted as in Fig D in S1 Text.

The validation accuracies reported in the main text were computed as the percent of validation data points that mapped onto the same trajectory ID as the original LOOPER computational scaffold over a specific time window. The time windows were chosen to align with their computational counterparts. Classifying F1 → mean accuracy between 1.39s and 3.19s, remembering F1 → minimum accuracy between 1.39s and 3.19s and comparing F2 to F1 → mean accuracy between 3.89s and 5.89s. The p-values were calculated by comparing the validation accuracy to a null model produced by randomly assigning trajectory IDs.

## 5 Analyses

Hidden Markov models were fit using the BNT toolbox and optimized using Bayesian Optimization. Parameters used for fitting can be found in Table 3. More details can be found in the S1 Text section **Hidden Markov Model Fitting**.

RNNs were trained with PyTorch using Adam with 10000 randomized samples. Noise was injected into the hidden and cell state of the LSTM units at each time step. More details can be found in the S1 Text section **Training of Recurrent Neural Networks**.

## Supporting information

**S1 Text. Detailed description of LOOPER method.** Contains details of all algorithms used in the LOOPER method, along with details of analysis used in the Main Text. Also includes Supplementary Figures.
(PDF)

## Acknowledgments

We thank Brenna Shortal, Drew Richardson for critically reading the manuscript. Human fMRI data were provided by the Human Connectome Project, WU-Minn Consortium (Principal Investigators: David Van Essen and Kamil Ugurbil; 1U54MH091657) funded by the 16 NIH Institutes and Centers that support the NIH Blueprint for Neuroscience Research; and by the McDonnell Center for Systems Neuroscience at Washington University.

## Author Contributions

**Conceptualization:** Connor Brennan, David Sussillo, Alex Proekt.

**Data curation:** Connor Brennan, Adeeti Aggarwal, Rui Pei, Alex Proekt.

**Formal analysis:** Connor Brennan, Alex Proekt.

**Funding acquisition:** Connor Brennan, Alex Proekt.

**Investigation:** Connor Brennan, Alex Proekt.

**Methodology:** Connor Brennan, Alex Proekt.

**Project administration:** Connor Brennan.

**Resources:** Connor Brennan.

**Software:** Connor Brennan.

**Supervision:** Connor Brennan.

**Validation:** Connor Brennan, Alex Proekt.

**Visualization:** Connor Brennan.

**Writing – original draft:** Connor Brennan, Alex Proekt.

**Writing – review & editing:** Connor Brennan, Alex Proekt.

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
