## [Decision Letter · Decision Letter 0]

15 Jul 2022

Dear Dr. Proekt,

Thank you very much for submitting your manuscript "One dimensional approximations of neuronal dynamics reveal computational strategy." for consideration at PLOS Computational Biology.

As with all papers reviewed by the journal, your manuscript was reviewed by members of the editorial board and by several independent reviewers. In light of the reviews (below this email), we would like to invite the resubmission of a significantly-revised version that takes into account the reviewers' comments.

We cannot make any decision about publication until we have seen the revised manuscript and your response to the reviewers' comments. Your revised manuscript is also likely to be sent to reviewers for further evaluation.

Sincerely,

Bard Ermentrout

Associate Editor

PLOS Computational Biology

Daniele Marinazzo

Deputy Editor

PLOS Computational Biology

Reviewer's Responses to Questions

**Comments to the Authors:**

Reviewer #1: This paper develops a method ("LOOPER") for extracting the core features of a neural population trajectory into a kind of "scaffolding" that can then be interpreted and used to compare the computational strategies used by very different systems. The method proceeds by constructing normalized diffusion maps from neuronal population time series data. This serves as a Markov model of the observed dynamics, whose states are then clustered and continuous trajectories are extracted. This method is successfully demonstrated on several interesting examples from a variety of systems.

I found the method and the results obtained to be quite interesting. The method is well-described and the results obtained across a diversity of systems were impressive. Thus, I support the publication of this paper with just a few minor comments for the authors to consider before finalizing their paper.

1) The terminology of "1-dimensional trajectories" used throughout the paper seems odd to me. Aren't all trajectories 1D objects?

2) What exactly does a "stable trajectory" mean (e.g., line 146 and elsewhere)? Stable w.r.t. what, exactly? Perturbations to the neuronal populations? Or multiple runs of LOOPER with different meta-parameter settings? Or something else?

3) I wonder if the authors could comment a bit more about the sensitivity of this method of analysis to the scale of the coarse-graining applied? It would seem that the particular scaffolding structure extracted by LOOPER could be heavily dependent on this scale.

4) Although the extracted scaffolds shown in the examples are certainly qualitatively very similar, I did notice differences in details such as the orders of transitions (e.g., see the Monkey/RNN comparisons in Figure 3). Are there any situations where this ordering might be important?

5) The references are incomplete. Some references are missing (e.g., the "?" in line 38), and the bibliography itself contains many references with incomplete citation information (e.g., refs 6, 12, 20, 43, 44, 50, 70, 72)

Reviewer #2: The paper proposes an algorithm (LOOPER) for reduction of large-scale neuronal recordings to a finite, countable set of one-dimensional trajectories (the so-called "reduced model") that presumably encode stimuli and how they are internally processed by the brain. LOOPER is simulated and tested for encoding and reconstruction of several types of neuronal datasets: 1) noisy recurrent neural networks (RNNs) trained to perform a working memory task, 2) mean firing rates of single neurons in primates’ prefrontal cortex during same working memory task, 3) calcium imaging data in C. elegans neurons, 4) visually evoked LFPs in mice, and 5) fMRI human data from a theory of mind task.

The algorithm seems very complicated, though thoroughly detailed in supplemental material. However, some of the algorithmic steps seemed to be added on the fly when the authors realized that they could not obtain the expected results (e.g., adding an "external" node to ensure completeness of loops in certain neuronal recordings), while others seemed so complicated that their interpretation is hard to understand (e.g., subsequent construction of several matrices and distances). Numerous minimization steps were also included, and it is not clear to me how they guarantee a valid solution. Description of LOOPER in the main text should at least address these points and clarify them.

It is not to say that LOOPER was not shown to successfully make some predictions. However, the results seem to apply only to bootstrapped and then trial-averaged data, and - apparently - not to the data themselves but rather to their projections on several principal components. A better description of the necessary steps on pre-processing of data used by LOOPER (including the justification) should be provided.

Other questions/concerns are the following:

1) Figure 3 and related results: it seems that the projection of neural activity of the first three principal components already shows the 6 trajectories that LOOPER finds in the end. Does this bias the algorithm to find (again) those 6 trajectories? I mean, what is the advantage of running such a complex algorithm for this example in which the 6 different trajectories were rather obvious from a simple principal component analysis anyway?

(Also, as a suggestion, it would be nice to keep same convention for solid lines versus dotted lines in panels A/C vs B/D)

2) if I interpret the results correctly (say in fig 3 but also the other sections) it seems that the biggest advantage of obtaining the reduced model (one-dimensional trajectories) with LOOPER is to identify timing of changes in neural activity that predict changes in behavior. The "timing" of these events (when trajectories either split or merge) seem important and, most probably, could not be determined otherwise -- neither in the complicated phase space of original trajectories (neural activity) nor in the space of PCA projections. I think the authors should emphasize more this property of the algorithm. This may also explain why the correlation values for BOLD data in Fig 5 are much lower compared to the other neural data which have better temporal resolution.

3) line 286 (and other places): "We then projected experimental observations from a non-overlapping validation dataset into the model space […]"

How exactly is this done?

Also, it is not clear to me how the reconstruction of data from the simplified model was done.

4) for some of the results, training and testing data depend on each other (e.g., Fig 4) raising questions about the strength of the reported results

5) a panel in Figure 5 is missing (Fig 5K)

6) there are several typos in the paper, including in the text of the figures themselves (e.g., Fig 4 "separation"; caption of Fig4, - p value; Fig 5 "initial trace"; etc). Some references are missing (line 38).

**Have the authors made all data and (if applicable) computational code underlying the findings in their manuscript fully available?**

Reviewer #1: **No: **The paper provides a very thorough description of each step of the method in the supplemental material. However, unless I missed it, I did not see any indication of where the data or code could be obtained.

Reviewer #2: **No: **It will be good to have the code of the algorithm posted on GitHub or some other code-sharing database.

PLOS authors have the option to publish the peer review history of their article (what does this mean?). If published, this will include your full peer review and any attached files.

Reviewer #1: No

Reviewer #2: No
---

## [Decision Letter · Decision Letter 1]

1 Dec 2022

Dear Dr. Proekt,

We are pleased to inform you that your manuscript 'One dimensional approximations of neuronal dynamics reveal computational strategy.' has been provisionally accepted for publication in PLOS Computational Biology.

Best regards,

Bard Ermentrout

Academic Editor

PLOS Computational Biology

Daniele Marinazzo

Section Editor

PLOS Computational Biology

Reviewer's Responses to Questions

**Comments to the Authors:**

Reviewer #2: The authors addressed my concerns adequately.

Note: There was no section in Methods about "Validation of computational scaffold" (page 92 of the pdf file, PCOMPBIOL-D-22-00958_R1), though it was cited in the text and in the response to reviewers.

I did see such section in the "clean" version of the file (page 44). Please make sure you do not forget to include it in the final draft of the manuscript.

**Have the authors made all data and (if applicable) computational code underlying the findings in their manuscript fully available?**

Reviewer #2: Yes

PLOS authors have the option to publish the peer review history of their article (what does this mean?). If published, this will include your full peer review and any attached files.

Reviewer #2: No

---

## [Editor Report · Acceptance letter]

18 Dec 2022

PCOMPBIOL-D-22-00958R1 

One dimensional approximations of neuronal dynamics reveal computational strategy.

Dear Dr Proekt,

I am pleased to inform you that your manuscript has been formally accepted for publication in PLOS Computational Biology. Your manuscript is now with our production department and you will be notified of the publication date in due course.

With kind regards,

Zsofia Freund
